# Extensive regulation of enzyme activity by phosphorylation in *Escherichia coli*

Evgeniya Schastnaya[1,2], Zrinka Raguz Nakic [1,2,3], Christoph H. Gruber [1,2], Peter Francis Doubleday [1], Aarti Krishnan[1], Nathan I. Johns[4,5], Jimin Park[4,5], Harris H. Wang[4,6] & Uwe Sauer[1✉]

Protein serine/threonine/tyrosine (S/T/Y) phosphorylation is an essential and frequent post-translational modification in eukaryotes, but historically has been considered less prevalent in bacteria because fewer proteins were found to be phosphorylated and most proteins were modified to a lower degree. Recent proteomics studies greatly expanded the phosphoproteome of *Escherichia coli* to more than 2000 phosphorylation sites (phosphosites), yet mechanisms of action were proposed for only six phosphosites and fitness effects were described for 38 phosphosites upon perturbation. By systematically characterizing functional relevance of S/T/Y phosphorylation in *E. coli* metabolism, we found 44 of the 52 mutated phosphosites to be functional based on growth phenotypes and intracellular metabolome profiles. By effectively doubling the number of known functional phosphosites, we provide evidence that protein phosphorylation is a major regulation process in bacterial metabolism. Combining in vitro and in vivo experiments, we demonstrate how single phosphosites modulate enzymatic activity and regulate metabolic fluxes in glycolysis, methylglyoxal bypass, acetate metabolism and the split between pentose phosphate and Entner-Doudoroff pathways through mechanisms that include shielding the substrate binding site, limiting structural dynamics, and disrupting interactions relevant for activity in vivo.

[1] Institute of Molecular Systems Biology, ETH Zurich, Zurich, Switzerland. [2] Life Science Zurich PhD Program on Systems Biology, Zurich, Switzerland. [3] Institute of Chemistry and Biotechnology, ZHAW Zurich University of Applied Sciences, Wädenswil, Switzerland. [4] Department of Systems Biology, Columbia University, New York, NY, USA. [5] Integrated Program in Cellular, Molecular, and Biomedical Studies, Columbia University, New York, NY, USA. [6] Department of Pathology and Cell Biology, Columbia University, New York, NY, USA. ✉email: sauer@imsb.biol.ethz.ch

Thriving in ever-changing environments, microbes constantly adapt their proteins by a multitude of mechanisms that range from slower transcriptional regulation to rapid modulation of protein activity through interaction with other proteins or small molecules. While most non-covalent regulatory interactions with proteins are fleeting in nature, covalent post-translational modifications (PTMs) can achieve long-term activity modulation that persists even after the initial stimulus has passed. Catalyzed by several hundred kinases and phosphatases[1,2], reversible serine/threonine/tyrosine (S/T/Y) phosphorylation is one of the most frequent PTMs, affecting up to 75% of all yeast or human proteins[3,4]. Although S/T/Y phosphorylation, and PTMs in general, are less abundant in prokaryotes and typically occur at a lower stoichiometry of modification[5], recent phosphoproteomic studies identified more than 2000 phosphorylation sites (phosphosites) on about 20% of the *Escherichia coli* proteins[6–11].

Thus, phosphoregulation may also be prevalent in bacteria, although only a few S/T- and Y-kinases and phosphatases are known and their in vivo substrates and regulators remain poorly characterized[12,13]. Mere detection of a phosphosite provides little evidence for function[14,15], which is typically inferred indirectly from phosphosite conservation, co-occurrence with other modifications, and correlation of the degree of protein phosphorylation with physiological variables such as metabolic flux[16–21]. Actual elucidation of function requires tedious in vitro phosphorylation of individual proteins and subsequent stability, activity, or interaction assays[22–25]. Consequently, less than 5% of the detected yeast phosphosites have a known function[26] and even fewer in *E. coli*[22,23,27,28]. At a larger scale, genetic perturbation of phosphosites has been coupled with phenotypic assays[21,29,30], the most recent of which demonstrated growth phenotypes for 42% of the 474 phospho-deficient yeast mutants under at least one of the 102 tested conditions[31]. The so far largest *E. coli* study mutated 134 PTM sites, including 48 phosphosites from a 2008 phosphoproteomics study[8], on enzymes at predicted regulation hotspots[30]. As may be expected from the prioritization, 88% of these acetylation and phosphorylation mutations affected fitness in at least one of seven tested conditions. In the meantime, the number of mapped *E. coli* phosphosites increased more than 20 times, thus the physiological role of the other 2000 reported sites remains opaque.

To assess more generally the functionality of S/T/Y phosphosites in bacteria, we focus here on *E. coli* central metabolism, 70% of whose enzymes were recently shown to be phosphorylated[8,30,32,33]. We mutated 52 reported phosphosites on 23 central enzymes to a non-phosphorylatable amino acid or a residue that mimics phosphorylation. Although neither enzymes nor sites were prioritized, 58% of the here investigated phosphosites caused a growth phenotype upon perturbation in at least one of two tested conditions. This high fraction of phenotypic consequences is surprising given that a similarly unbiased screen in yeast had to test more than a hundred conditions to identify phenotypes in 42% of the mutants[31]. Determining metabolic profiles of our phosphomutants provided further indication of the functionality, even in the absence of a phenotype. Overall, we present evidence of functionality for 44 of the 52 investigated phosphosites, suggesting an extensive role for regulatory phosphorylation in *E. coli* metabolism. By combining in vitro and in vivo experiments for selected cases, we demonstrate how single phosphosites modulate enzymatic activity and regulate metabolic fluxes in glycolysis, methylglyoxal bypass, acetate metabolism and the split between pentose phosphate (PP) and Entner–Doudoroff (ED) pathways.

## Results

### Most phosphosite mutations affect growth physiology.
To systematically explore the functional relevance of phosphorylation in *E. coli*, we focused on phosphosites that were reported in at least two of the six available phosphoproteomics studies[6–11]. Specifically, we selected 52 single phosphosites or multiple phosphosites in close proximity located on 23 enzymes, including transferases, isomerases, oxidoreductases, and lyases (Fig. 1a and Supplementary Data 1). Only four enzymes were located outside of central metabolism in nucleotide, lipid, and amino acid metabolism and oxidative stress pathways. The phosphorylated residues were mutated by multiplex automated genome engineering (MAGE)[34] to other natural amino acids that mimic the non-phosphorylated or the constitutively phosphorylated state, corresponding to 0% and 100% phosphorylation stoichiometry, respectively[35,36]. While these mutations are pertinent to assess the overall functionality of a phosphosite, they do not necessarily reflect the site occupancy in vivo, which was reported for only nine phosphosites investigated in this study (Supplementary Data 1). To abolish phosphorylation, phosphorylatable hydroxy groups were removed by mutating S and T to alanine (A), and Y to phenylalanine (F). To mimic phosphorylation, the negative charge of phosphorylation was imitated by substituting S and T with glutamic acid (E). Phosphorylation of Y cannot be mimicked due to the lack of a suitable natural amino acid. Ten of the 52 phosphosites were previously suggested to be functionally relevant, including S113 of isocitrate dehydrogenase (Icd), the paradigm for phosphorylation in *E. coli*[22,30].

To assess phosphosite functionality, we first determined physiological phenotypes in microtiter plate growth assays. As a baseline condition, all mutants were grown in a minimal medium with glucose. Additionally, individual mutants were grown on either acetate, pyruvate, glycerol, or fructose to maximize variation in fluxes through the targeted reactions[37]. We also included deletion mutants of the investigated enzymes for comparison to complete loss of function[38]. Overall, mutations of 30 out of 52 phosphosites affected the growth rate under at least one condition (difference to wild-type growth rate ≥ 10%, *p*-value < 0.05) (Fig. 1b,c and Supplementary Data 2). On average, growth defects were less pronounced in phosphomutants than in deletion strains. A similar distribution of growth effects in phosphomimetic and abolishing mutations suggested that phosphorylation can be both activating and inhibiting (Fig. 1b). Outside of central metabolism we observed mild growth defects for phosphomutants in lipid synthesis (KdsD) and oxidative stress pathway (AhpC), and no phenotypes for Adk and MetK phosphomutants, even though the MetK phosphomimetic mutant was preferred on glucose in a competition screen[30]. Comparison of phosphomutant growth profiles to the respective deletion strains suggested that phosphorylation activates ManX and Gnd, inhibits PykA and TpiA, and has site-dependent effects on GpmA, PykF, SucB, and AceF. Phosphosite mutations in Icd, AcnB and Pta caused condition-specific lethality, as in these cases catalytic residues were mutated. Overall, the large fraction of growth defects in metabolic phosphomutants suggests an important and so far underappreciated role of phosphoregulation in coordinating central metabolism.

### Metabolomics revealed phosphosite functionality without growth phenotypes.
While altered growth phenotypes strongly suggest functionality for the majority of investigated phosphosites, the absence of growth phenotypes does not exclude the possibility of metabolic compensation in response to altered enzyme activity. To assess functional consequences of mutations, we determined the intracellular metabolome by flow-injection analysis time-of-flight mass spectrometry (FIA TOF-MS)[39] of all mutants because it can detect compensation such as local changes in substrate and/or product levels of mutated enzymes due to altered kinetics or flux rerouting, as has been shown

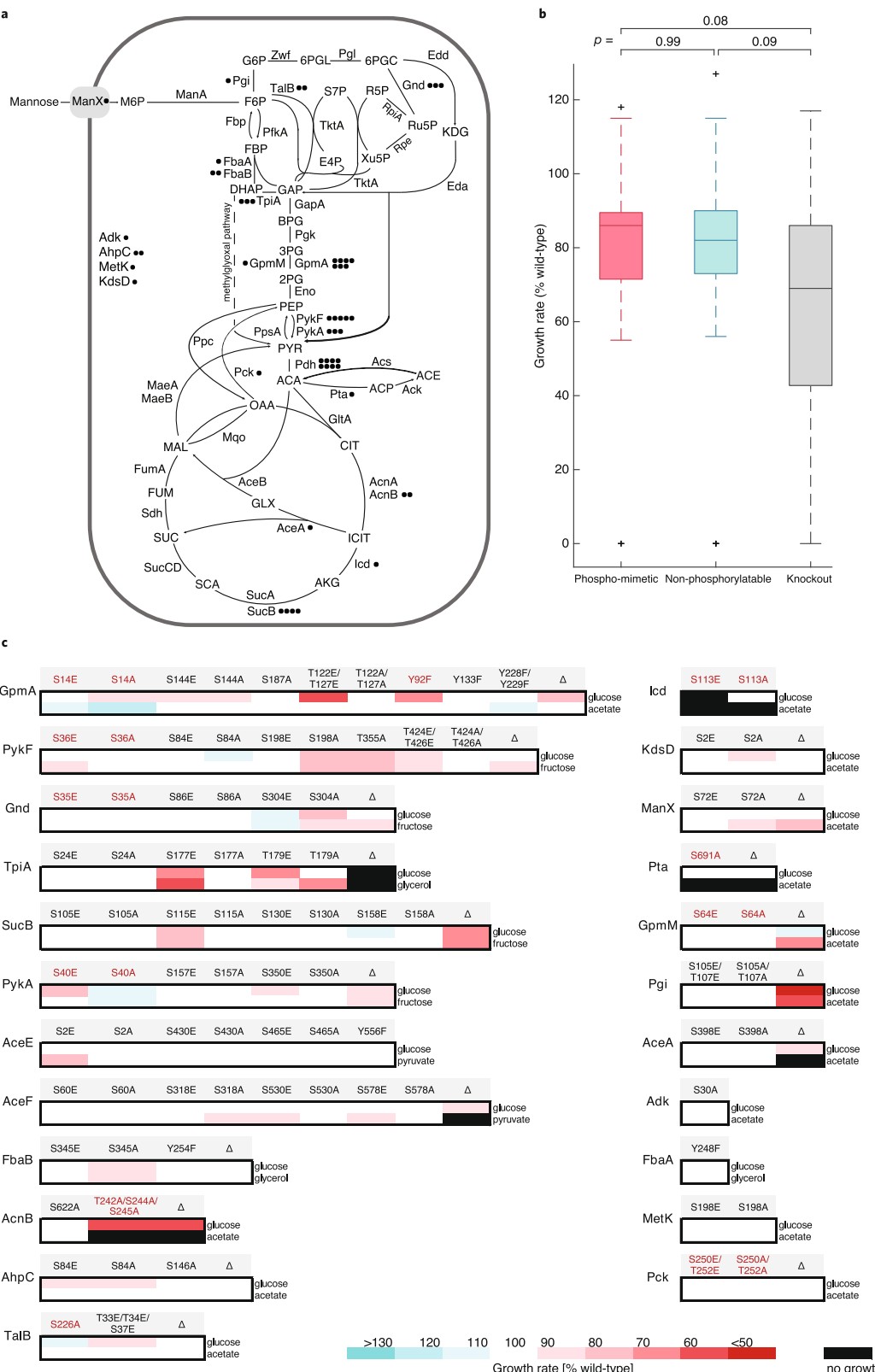

previously[29,40]. Metabolite extracts were obtained from exponentially growing cultures on the same carbon sources used for phenotypic screening. Ions detected in the 50-1000 Dalton range were annotated to 460-500 metabolites using a genome-wide model of *E. coli*[41]. Metabolite changes with an absolute log₂ fold change > 0.38 (95% percentile of the fold change distribution for all annotated ions in all strains) and a Benjamini–Hochberg corrected $p$-value < 0.05 were considered significant.

The majority of the 89 phosphomutants analyzed exhibited metabolic changes in at least one of the tested conditions (Supplementary Data 3). While phosphomutants exhibited generally similar metabolic responses in both conditions with a

**Fig. 1 Localization and growth effects of phosphosite mutations. a** Localization of investigated phosphomutants in central metabolism of *E. coli*. Circles represent the number of mutated phosphosites for a given enzyme. Pdh—pyruvate dehydrogenase complex, the phosphosites were mutated on its AceE and AceF components. **b** Distribution of significantly changing growth rates of mutants in all conditions. Significance cutoff: growth rate change ≥ 10% compared to the wild-type, $p < 0.05$. Box's length represents the interquartile range (IQR) (25th to 75th percentiles), the central mark shows the median, the whiskers span 1.5 times IQR, the outliers are plotted individually. The number of strains used to calculate the distributions was 24 for phosphomimetic, 26 for non-phosphorylatable, and 19 for knockout mutants. Statistical significance was calculated with a two-tailed unpaired *t*-test and *p*-values are indicated. **c** Relative growth rates of phosphomutants and corresponding knockout strains in minimal medium with the indicated carbon sources. Mutants of residues with catalytic functions proposed by similarity or structural studies are shown in red.

median of four changing metabolites, some phosphosite mutations had more substantial consequences when flux through the reaction was higher than on glucose; i.e. the number of changing metabolites increased at least by 30 for *AceE S2E*, *TpiA S177E*, and *AceA S398A* (Supplementary Data 3). The metabolic consequence of a phosphosite mutation provided evidence for function when either phosphomimetic or abolishing mutant exhibited: (i) local metabolic rearrangements, defined as significant changes in the abundance of reactants or metabolites within five reaction steps from the mutated enzyme (Supplementary Data 4), and (ii) correlation to the metabolic profile of the corresponding gene knockout (Spearman correlation coefficient > 0.4, *p*-value < 0.001) (Fig. 2a and Supplementary Data 5) in any of the tested growth conditions. Assuming that (de)phosphorylation can also inhibit enzymatic activity, the metabolic response to a phosphosite perturbation and gene knockout should be similar. The majority of the observed correlations between mutant strains were mild (Spearman correlation coefficient < 0.4) and not specific to phosphosite perturbation directionality (Fig. 2a). Local metabolic changes and concordant metabolic signatures between a phosphomutant and a knockout provided evidence of functionality for 38 phosphosites, including 14 that were phenotypically silent (Fig. 2b). By comparing the phenotypic and metabolic outcomes of abolishing or mimicking phosphorylation, and where available the enzyme deletion, we hypothesized positive or negative consequences of phosphorylation on enzymatic activity for the majority of the investigated sites (Fig. 2b). Only 13 phosphosites did not exhibit a metabolic phenotype, eight of which also did not have a growth phenotype. Overall, combined metabolic and phenotypic evidence suggested functionality for 85% of the investigated phosphosites.

**In vitro characterization of phosphosite mutations.** To validate functionality, we next characterized in vitro properties of enzymes from phosphosite mutants with at least 25% reduced growth rate compared to wild-type; i.e. AcnB T242/S244/S245, Pta S691, SucB S115, GpmA S14, Y92, T122/T127, PykF S198, TpiA S177 and T179. Additionally, we included Gnd S304 that displayed opposing growth effects for mimicking and abolishing mutations. These phosphosite mutations were introduced into the seven genes encoded on ASKA library plasmids with His-tags[42]. To exclude protein misfolding as the cause of mutant phenotypes, we performed thermal shift assays with wild-type and phosphomutant enzymes purified from *E. coli* overexpression strains (Supplementary Data 6). Phosphomutants of Gnd, PykF, TpiA, Pta, and SucB exhibited no or below 1 °C change in melting temperature ($T_m$), demonstrating no significant changes in folding (Fig. 3a). For GpmA and AcnB we observed considerably decreased $T_m$, unexpectedly, also in the abolishing mutants mimicking the unphosphorylated wild-type enzyme, suggesting that any mutation of the investigated residues can affect protein folding and stability. While this does not rule out the functionality of these phosphosites, the observed growth phenotypes and metabolic changes for *GpmA S14*, *Y92*, *T122/T127*, and *AcnB T242/S244/S245* phosphomutants were probably caused by

misfolded proteins. For *AcnB T242/S244/S245* it is tempting to speculate that phosphorylation is likely inhibitory, as S244 and S245 are substrate-binding residues[43]. For further validation, we focused on the enzymes with unchanged $T_m$, except SucB that functions only as part of the 2-oxoglutarate dehydrogenase complex. Nevertheless, we concluded that phosphorylation at S115 has an inhibitory effect on the SucB reaction based on growth inhibition, metabolic correlation with the knockout, strong accumulation of substrate, no change in $T_m$ in the mimicking *SucB S115E* mutant and almost no effects in the abolishing *SucB S115A* mutant (Figs. 1c, 2a, 3a, b).

To assess whether the physiological consequences of phosphosite mutations were indeed caused by altered enzyme activities, we determined in vitro activities for TpiA, Pta, Gnd and PykF phosphomutants (Fig. 3c and Supplementary Data 7). Phosphomimetic mutation of TpiA S177E resulted in a barely active enzyme, while TpiA T179E had milder effects, consistent with the more extreme changes in neighboring metabolite abundances seen in vivo for these mutations (Supplementary Data 3). In accordance with our previous hypothesis, mutating the catalytic residue S691 of Pta to mimic or abolish phosphorylation reduced enzyme activity significantly, likely by interfering with substrate binding (Fig. 3c). While phosphoinhibition of Pta S691 is most likely achieved through interference with the active site[44], phosphoinhibition of TpiA at S177 and T179 is presumably allosteric in nature by hindering the structural dynamics required for catalysis[45]. The physiological and metabolic data suggests that phosphorylation of Gnd S304 and PykF S198 activates enzyme activity (Fig. 2b), but the in vitro activities of these mutants were indistinguishable from their wild-type counterparts (Fig. 3c).

**Phosphorylation directly inhibits Pta and TpiA, while the in vivo effects on Gnd and PykF activity are indirect.** Does the above identified phosphoregulation control in vivo pathway usage? During growth on glucose, inhibition of TpiA via phosphomimetic mutation increased the concentration of the methylglyoxal pathway intermediate (R)-S-lactoylglutathione (Fig. 3d). The site occupancy of S177 and T179 was recently reported to be around 15% during growth on glucose[13], a percentage that significantly decreases TpiA in vitro activity (Supplementary Fig. 1) and hence glycolytic flux. Since phosphorylation at S177 directly inhibits TpiA activity, likely via limiting the crucial loop-6 movement upon substrate binding[45], we provide evidence for a flux redirection from glycolysis to the methylglyoxal pathway during glucose catabolism. Inhibition via phosphomimetic mutation at T179 has a similar effect but since the inhibition of in vitro enzyme activity was much lower (Fig. 3c), also the in vivo consequences are milder (Fig. 3d). To prove that the phosphomimetic Pta S691E is indeed inactive in vivo, the inactive genomic *Pta S691A* mutant was supplemented with either a wild-type or S691E Pta expressing plasmid, demonstrating that only the wild-type but not the phosphomimetic Pta enzyme could rescue S691A lethality on acetate (Fig. 3e). Overall, our results suggest that acetate secretion can be regulated via inhibitory phosphorylation.

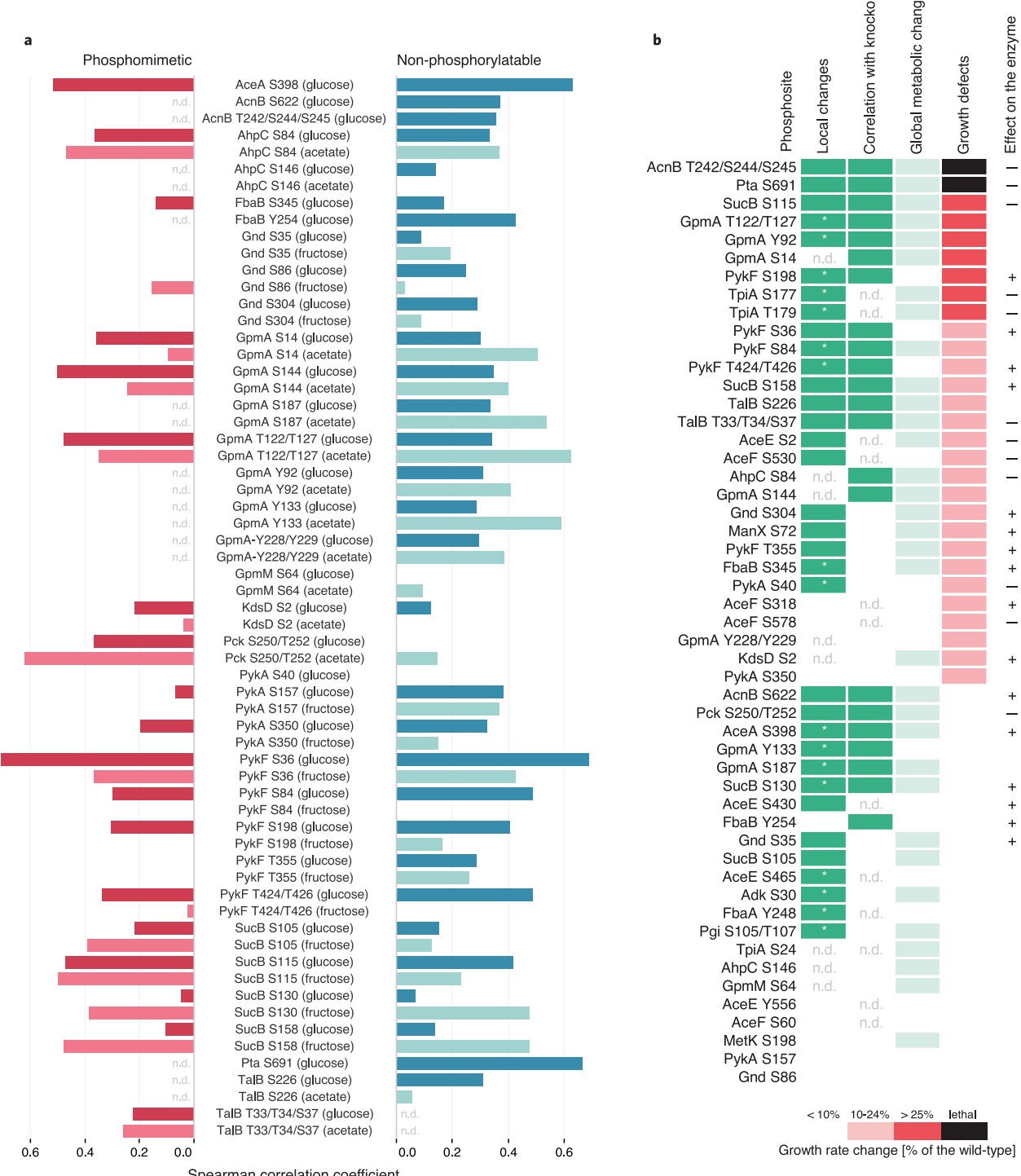

**Fig. 2 Metabolic evidence for phosphosite functionality. a** Spearman correlation of metabolic profiles of phosphomimetic (red) and non-phosphorylatable (petrol) phosphomutants with the corresponding gene knockout. Bright and shaded colors represent glucose and alternative growth conditions, respectively. Negative correlation values are not shown. N.d.—not determined. **b** Functionality ranking of mutated phosphosites based on phenotypic and metabolic evidence. Metabolic evidence, shown in green, is defined as local changes in reactants or metabolites within five reaction steps from the mutated enzyme (*), and correlation to the metabolic profile of the corresponding gene knockout. Albeit not used for ranking, global metabolic changes defined as at least eight metabolites (twice the median amount of changing metabolites for all phosphomutants in all conditions) are indicated in light green. Where possible, negative (–) or positive (+) effects of phosphorylation is concluded based on the phenotypic and metabolic evidence for the phosphomutants. N.d.—not detectable.

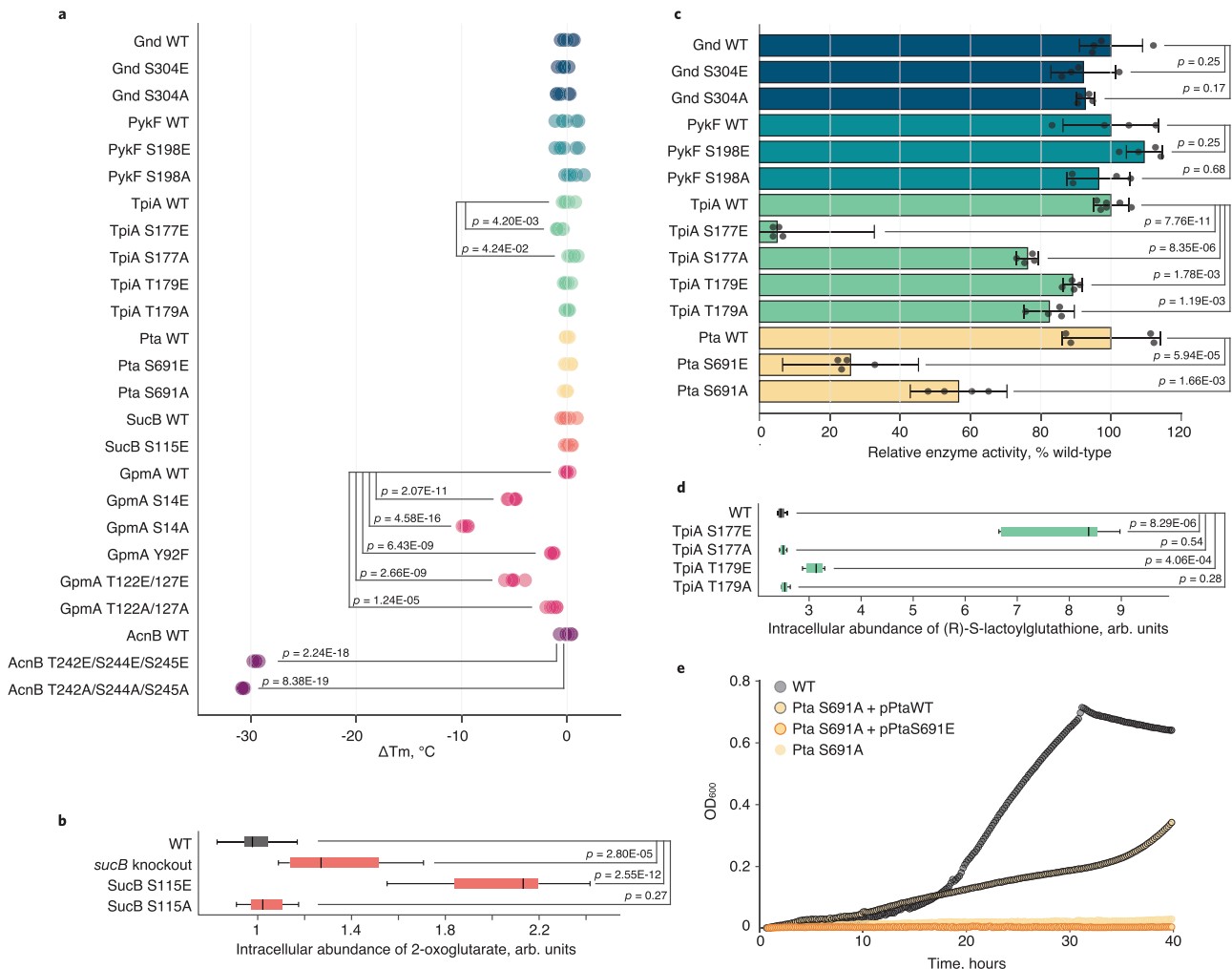

**Fig. 3 Substantial effects of phosphosite perturbation. a** Change in thermal stability of purified (phosphomutant) enzymes. For each of the six replicates, the difference between its melting temperature and the average melting temperature of the respective wild-type enzyme ($\Delta T_m$) is plotted. **b** Intracellular abundance of 2-oxoglutarate in phosphomutants of SucB S115, the *sucB* knockout and the wild-type *E. coli* during growth on minimal medium with fructose. $n = 10, 3, 4, 5$ biological replicates with two technical replicates were measured for the wild-type, knockout, S115E, S115A mutants, respectively. **c** Relative activity of purified wild-type and phosphomutant enzymes, determined as the maximum initial reaction rate. Data represent the mean ± standard deviation. The following substrate concentrations were used: Gnd—5 mM 6-phosphogluconate, 1 mM NADP$^+$; PykF—2 mM phosphoenolpyruvate, 1 mM ADP; TpiA—1 mM dihydroxyacetone phosphate; Pta—5 mM acetyl-phosphate, 1 mM coenzyme A. Four independent replicates (six for TpiA wt) were measured. **d** Intracellular abundance of the methylglyoxal pathway intermediate (R)-S-lactoylglutathione in phosphomutants of TpiA S177, TpiA T179, and wild-type *E. coli* during growth on minimal medium with glucose. $n = 3$ biological replicates with two technical replicates were measured for each strain. For (**a**–**d**) statistical significance was calculated with a two-tailed unpaired *t*-test and *p*-values (in (**b**, **d**) Benjamini–Hochberg adjusted) are indicated. Box's length in (**b**, **d**) represents the IQR, the central mark shows the median, the whiskers span 1.5 times IQR. **e** Rescue of Pta S691A lethality on minimal medium with acetate. Acetate growth curves for wild-type, Pta S691A, and Pta S691A containing the plasmid carrying either wild-type Pta, or Pta S691E are shown. Source data are provided as a Source Data file.

Data interpretation was straightforward for TpiA and Pta with altered in vitro activities, leaving us with Gnd and PykF. In both cases, growth rates of phosphoabolishing mutants were reduced (Fig. 1c), yet in vitro activity of the purified mutant enzymes was unaltered (Fig. 3c). Since the melting temperatures of the mutants were effectively unchanged (Fig. 3a), misfolding is likely not responsible for the observed growth defects. Moreover, we found no difference in PTMs between overexpressed wild-type and mutant enzymes, suggesting absence of compensatory modifications (Supplementary Fig. 2). Since Gnd and PykF function in vivo as a homodimer and homotetramer, respectively[46,47], we next checked whether phosphosite mutations affected oligomerization of overexpressed enzymes. While the majority occurred as monomers, the ratio of monomers to multimers was unaltered in the phosphomutants (Supplementary Fig. 3), suggesting that

phosphorylation at these non-catalytic residues is not required for oligomerization. Thus, the most parsimonious explanation is that phosphorylation allosterically modulates in vivo activity of the multimeric forms. Alternatively, phosphorylation might be relevant for other in vivo properties, such as preventing aggregation during exponential growth as in the case of yeast pyruvate kinase[48].

Phenotypic similarity between *gnd* knockout and the abolishing Gnd S304A mutant (Fig. 2b) suggests that phosphorylation is necessary for in vivo flux through the oxidative part of the PP pathway at the 6-phosphogluconate branch point to the ED pathway. Although reaction substrate and product levels were unaltered in knockout and Gnd S304A mutant, the phosphomimetic mutant Gnd S304E not only grew faster than the wild-type (Fig. 1c) but also had almost 30% lower reaction substrate levels

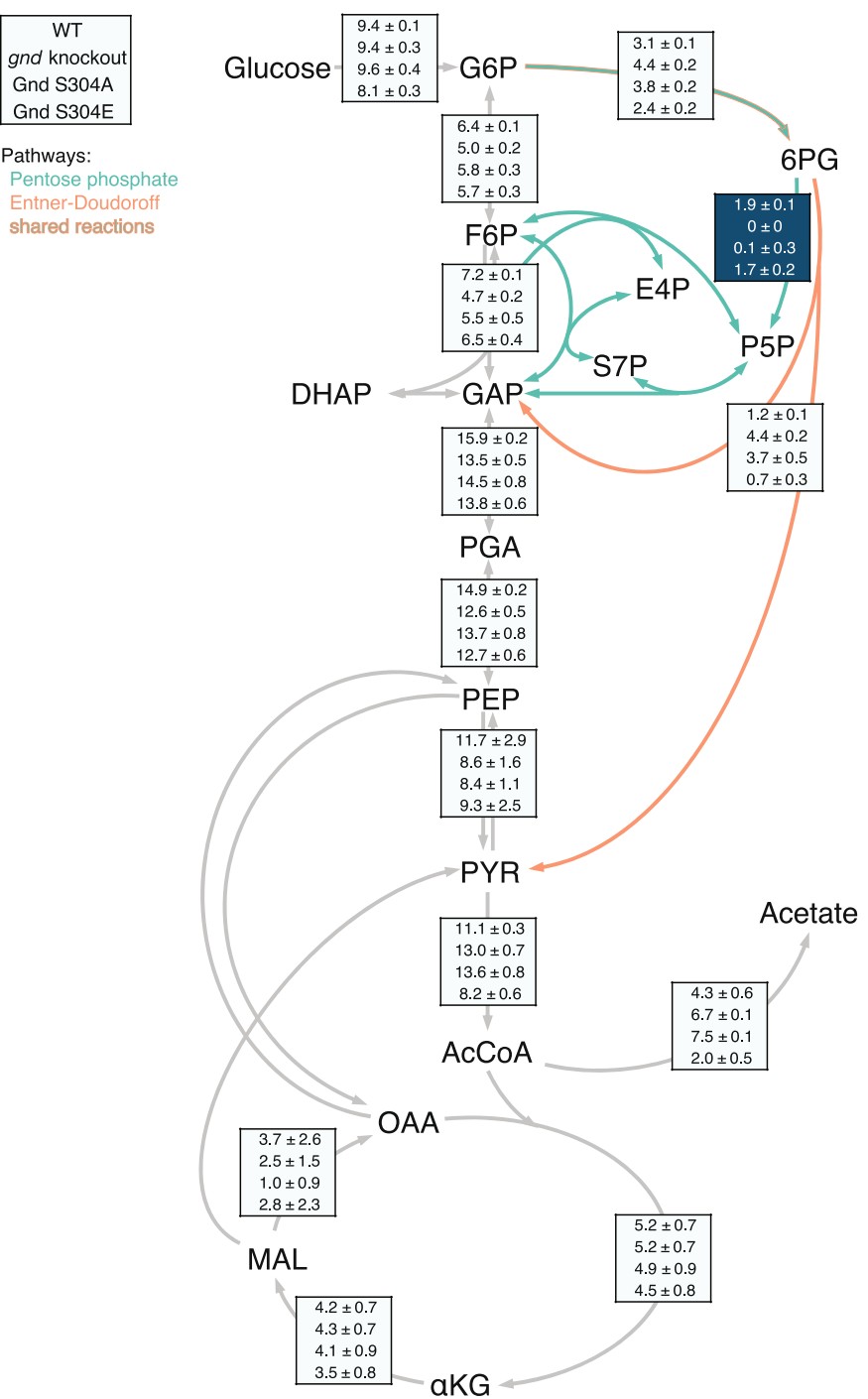

**Fig. 4 Absolute intracellular fluxes in Gnd knockout and phosphomutants.** $^{13}$C-based flux map of the wild-type (first), *gnd* knockout (second), phosphoabolishing *gnd* S304A (third), and phosphomimetic *gnd* S304E (fourth) mutants during exponential growth on glucose. Flux through the Gnd-catalyzed reaction is highlighted in dark blue. Flux values are given in mmol g CDW$^{-1}$ h$^{-1}$. Source data are provided as a Source Data file.

(*p*-value = 0.015) during growth on fructose (Supplementary Data 3), providing further evidence that phosphorylation at S304 increases Gnd activity in vivo. To validate in vivo activation of Gnd through phosphorylation, we estimated intracellular fluxes by $^{13}$C-constrained flux balancing from growth experiments on 100% [1-$^{13}$C]glucose and a mixture of 20% [U-$^{13}$C] with 80% natural abundance glucose[49,50]. As expected, the wild-type catabolized glucose mainly through glycolysis and the PP pathway with lower ED pathway flux[50] (Fig. 4). Deletion of *gnd* blocked utilization of 6-phosphogluconate through the PP pathway and rerouted flux into the ED pathway, as was described

before[51–53]. Consistent with the hypothesis that phosphorylation at S304 activates Gnd, fluxes in the phosphoabolishing Gnd S304A mutant were similar to the knockout with an active ED pathway and almost no flux through the Gnd reaction, while fluxes in the phosphomimetic Gnd S304E were similar to the wild-type (Fig. 4). In vivo phosphorylation at S304 of Gnd was identified with 100% localization probability in four out of six *E. coli* phosphoproteomics studies[6,7,9,11]. Thus, even though phosphorylation at non-catalytic S304 does not change in vitro Gnd activity, it is crucial for in vivo activity and regulates the PP pathway flux.

## Discussion

Compared to eukaryotes, fewer bacterial proteins are phosphorylated and at a lower stoichiometry, hence S/T/Y phosphorylation was historically considered less relevant in prokaryotes[5]. Despite the recent expansion to more than 2000 detected phosphosites in *E. coli*[6–11], phosphoregulation is still only demonstrated for few cases, excluding autophosphorylation; i.e. S113 of Icd, S226 of TalB, S372 of enolase, Y71 of UDP-glucose dehydrogenase, T419 of phosphoenolpyruvate synthase, and S239 of glutamate-tRNA ligase[22,30,54–56]. An additional 38 phosphosites on enzymes at key flux-regulating nodes were reported to affect fitness upon perturbation[30]. Here, we systematically characterized the functionality of S/T/Y phosphorylation in *E. coli* metabolism by analyzing phenotypic and metabolic consequences of mimicking or abolishing phosphorylation at 52 phosphosites on 23 central enzymes. In sharp contrast to eukaryotes[29,57–60], the surprising outcome is that about 85% of the modified sites cause altered phenotypes and/or metabolic rearrangements, providing strong evidence of functionality. In some cases, the consequences of single, non-catalytic phosphosite mutations were as drastic as a more than 40% growth rate reduction. Since serine and threonine are also subject to other modifications, we cannot exclude that some phenotypic changes following replacement with alanine might at least in part be influenced by other PTMs. Overall, our results demonstrate that the fraction of functional phosphosites is higher in *E. coli* than in eukaryotes[57–60], and/or they have a more direct effect on the phenotype.

The in vivo occupancy of phosphosites varies with conditions and is typically lower in prokaryotes[5], with a median occupancy of 6.7% in *E. coli* during the here used condition of growth on glucose[9]. Based on the phenotypic and metabolic consequences of turning phosphosites "off" or "on" by mimicking 0 or 100% occupancy, we show that phosphorylation has a positive effect on the activity of PykF, Gnd, ManX, FbaB, KdsD, and AceA, an inhibitory effect on TpiA, AhpC, PykA, and Pck, and opposite effects at different phosphosites of SucB, AceF and AceE. For AcnB and Pta, inhibition occurs probably through phosphorylation of catalytic residues. Since Pta is likely to be a structural homolog of Icd whose substrate-binding is inhibited by phosphorylation through the kinase AceK[44,61], it is tempting to speculate that acetate node and TCA cycle flux are both inhibited by AceK to minimize futile cycling in acetate utilization and subsequent use of the glyoxylate shunt, by blocking the Pta and Icd reactions, respectively. While some phosphosite mutations affect cellular fitness by causing misfolded proteins (i.e. GpmA and AcnB), unaltered thermal stability of five of the seven mutated enzymes suggests their phenotypes to originate from modulated protein activity rather than misfolding. We demonstrated reduced glycolytic flux and increased activity of the methylglyoxal pathway upon allosteric phosphoinhibition of TpiA at S177 and T179, likely by limiting the structural movement required for catalysis. The recent demonstration of increased phosphorylation at S177 and T179 in a YeaG kinase knockout indicates that YeaG downregulates another kinase that phosphorylates TpiA[13]. For PykF and Gnd, our results suggest more complex phosphoregulation such as promoting protein-protein or protein-metabolite interactions, since phosphorylation was required in vivo but did not affect in vitro enzyme activity. For Gnd, [13]C-flux analysis demonstrated that phosphorylation at S304 was required for in vivo PP pathway flux because preventing phosphorylation caused knockout-like fluxes and phenotype. The fact that only phosphorylated Gnd is active in vivo may contribute to previous observations that the PP pathway operates below its maximum in vitro capacity[62]. Abolishing phosphorylation at S304 effectively redirects the flux from the PP to the ED pathway.

As 36 of the 44 here identified functional phosphosites were never described before, we nearly double the number of phosphosites with known regulatory function in *E. coli*. We provide evidence that protein phosphorylation is a major regulation network in bacterial metabolism, capable of controlling metabolic fluxes by various mechanisms, such as shielding the substrate binding site, allosterically limiting structural dynamics, and disrupting interactions relevant for in vivo activity. Since only eight S/T- and Y-protein kinases are known in *E. coli*, i.e. AceK, PpsR, HipA, YeaG, SrkA, YegI, Wzc, Etk[7,13,22,23,54,63–65], our results suggest that *E. coli* kinases are either less specific than eukaryotic counterparts or many kinases remain hidden in the genome.

## Methods

**Strain construction**. Genomic point mutations of *E. coli* phosphosites were constructed using MAGE[34]. We designed 90 bp lagging-strand targeting ssDNA oligos to introduce phosphosite mutations (Supplementary Data 8). To abolish phosphorylation, serine and threonine were mutated to alanine, and tyrosine to phenylalanine; to mimic phosphorylation, serine and threonine were substituted with glutamic acid. Previously described oligo design strategies, including four phosphorothioated bases at the 5′ terminus of the oligo, were used to optimize incorporation efficiency[66]. *E. coli* MG1655 Δ*mutS* with kanamycin resistance (referred to as wild-type) harboring the temperature-sensitive λ-Red recombineering plasmid pSIM5[67] with chloramphenicol resistance was grown overnight in a shaker at 30 °C in LB-Lennox medium (10 g/L tryptone, 5 g/L yeast extract, 5 g/L NaCl) with 25 μg/mL chloramphenicol. The next day, 100 μL of overnight culture was transferred into 3 mL of fresh LB-Lennox medium with 25 μg/mL chloramphenicol and grown under shaking at 30 °C until an optical density at 600 nm ($OD_{600}$) of 0.5–0.8. The λ-Red proteins were induced by transferring the cultures into 42 °C for 15 min, the cultures were put on ice immediately afterwards. 1 mL aliquots were washed twice through centrifugation and resuspension in 1 mL of ice-cold d$H_2O$. Final washed pellets were resuspended in 50 μL of 2.5 μM oligo. Cell-oligo mixture was transferred in a prechilled 0.1 cm gap electroporation cuvette (Bio-Rad) and electroporated using the following settings: 1.5 kV, 200 Ω, 25 μF (Bio-Rad Gene Pulser). After electroporation, 1 mL of LB-Lennox was immediately added and transferred to a tube containing 2 mL of LB-Lennox without antibiotics. After an hour of recovery at 30 °C, 25 μg/mL chloramphenicol were added, cultures grown until $OD_{600}$ 0.5-0.8, and MAGE was repeated. Overall, 3–4 MAGE cycles were performed for each mutation. After the last MAGE cycle, cultures were grown for 3–4 h, streaked on LB agar plates and incubated at 30 °C overnight. Mutant colonies were identified using MASC-PCR[66] and sequenced to confirm the mutation. The mutants were grown at 42 °C for 4–5 h in LB-Lennox without antibiotics and at 37 °C on LB agar plates overnight to cure the pSIM5 plasmid. *E. coli* gene deletion mutants were retrieved from the KEIO collection[38].

**Microplate cultivation and growth analysis**. *E. coli* cells were grown in M9 minimal medium supplemented with either 5 g/L glucose, 5 g/L fructose, 6.8 g/L sodium acetate, 6.1 g/L sodium pyruvate or 5.1 g/L glycerol to keep C-atoms constant between the different carbon sources. Precultures in deep 96-well plates were inoculated from glycerol stocks in LB medium with 50 μg/mL kanamycin and grown overnight. Dilution rows of LB precultures were inoculated for a second preculture in 800 μL of M9 medium with 5 g/L glucose and grown for 5–6 h in deep 96-well plates. Cultures at an $OD_{600}$ 0.5–0.8 were used to inoculate flat-bottom 96-well microplates (ThermoFisher, Cat. # 167008). In each well, cultures of 200 μL were inoculated with a staring $OD_{600}$ of 0.01–0.03, the plate was sealed with parafilm and grown at 37 °C and 880 rpm in a microplate reader (Tecan Infinite M Nano). $OD_{600}$ was recorded every 10 min for 18–40 h using the Tecan i-control 3.9.1.0 software. Each mutant and the wild-type were grown in triplicates and the experiment was repeated on a different day, resulting in six growth curves for each mutant. A linear fit on the log-transformed growth curve was used to determine the growth rate. Growth of mutants was compared to growth of the wild-type reference strain using a two-sided 2-sample *t*-test assuming unequal variance, and growth rates with a *p*-value < 0.05 and >10% difference from the wild-type were considered significant.

**Metabolite extraction**. *E. coli* strains were grown in 1 mL M9 minimal medium supplemented with the same carbon sources used for growth analysis. Cultures were grown in deep 96-well plates at 37 °C and 250 rpm until the $OD_{600}$ of 0.4–1.5 was reached (mid-exponential phase). For extraction, the plate was centrifuged for 2 min at 0 °C and 2250 × *g* and the supernatant discarded. For cold extraction, 150 μL of −20 °C pre-cooled 40% (v/v) acetonitrile, 40% methanol, 20% water solution was added into each well containing a cell pellet and incubated at −20 °C for 1 h. For hot extraction, 100 μl of 80 °C 60% (v/v) ethanol buffered with 10 mM

ammonium acetate at pH 7.5 was added to the pellet and the plates were incubated for 3 min at 80 °C with three vortexing steps. After extraction, plates were centrifuged for 5 min at 0 °C and 2250 × g and supernatant stored at −80 °C until measurement. Metabolic extracts of AcnB, Adk, AhpC, GpmM, KdsD, ManX, MetK, Pck, Pgi, Pta, and TalB mutants were prepared using the hot extraction procedure, and for other mutants cold extraction was used.

**FIA TOF-MS measurement and untargeted metabolomics data processing.** Metabolite extracts were analyzed by direct flow double injection on an Agilent 6550 series iFunnel quadrupole time-of-flight mass spectrometer (Agilent, Santa Clara, CA, U.S.A.) coupled to a GERSTEL MPS2 autosampler[39]. A sample volume of 5 μL was injected into a constant flow of isopropanol/water (60:40, v/v) buffered with 5 mM ammonium carbonate (pH 9), containing 3-Amino-1-propanesulfonic acid (138.0230374 m/z, Sigma Aldrich) and hexakis(1H,1H,3H-tetra-fluoropropoxy)phosphazine (940.0003763 m/z, HP-0921, Agilent Technologies) for online mass axis correction. The ion source parameters were set as follows: 225 °C source temperature, 11 L/min drying gas, 20 psig nebulizer pressure, and TOF settings as follows: 350 V fragmentor voltage, 750 V octopole voltage. Mass spectra were recorded in negative ionization mode within a mass/charge ratio range of 50–1000 m/z using the highest resolving power (4 GHz HiRes) with an acquisition rate of 1.4 spectra per second. Mass spectrometry data processing and analysis were performed in Matlab (The Mathworks, Natick). Deprotonated ions were annotated based on mass using 0.001 Da tolerance using a genome-wide reconstruction model of *E. coli* metabolism[41]. For every ion, the abundances from all replicates of a given mutant were pooled and compared to the pooled abundances of the wild-type sample. The $\log_2$ fold change of an ion abundance in the mutant compared to the wild-type was determined, and a two-sided 2-sample *t*-test with unequal variance was performed. The obtained *p*-values were corrected for multiple testing using the Benjamini–Hochberg procedure. A fold change cutoff that removes very small changes was chosen based on the 95% of the distribution of the fold changes of all annotated ions in all strains. Ions with a $\log_2$ fold change > ±0.38 and a corrected *p*-value < 0.05 were considered significantly changing. For network distance analysis, MetaboSignal R Package was used[68] and metabolites with MS_distances ≤ 5 were considered as a local network. Metabolites that are not included in the *E. coli* KEGG network were curated manually. To determine the similarity between different mutants, pairwise Spearman's rank correlation coefficient was calculated based on the fold changes of all annotated ions in one strain versus another.

**Expression and purification of phosphomutant enzymes.** Back-to-back primers carrying phosphosite mutations (Supplementary Data 8) were used for site-directed mutagenesis of ASKA library plasmids without GFP[42]. The phosphomutant plasmids were introduced into chemically competent *E. coli* BL21 and verified by sequencing. Wild-type and phosphomutant enzymes were overexpressed by cultivating the strains with respective plasmids overnight in 200 mL of LB medium with 25 μg/mL chloramphenicol and 0.1 mM IPTG. Cell pellets were obtained by centrifugation, lysed and centrifuged to obtain the protein-containing supernatants, which were loaded on His GraviTrap™ Talon® (GE Healthcare) and the His-tagged fraction was eluted with buffer B containing 500 mM imidazole. The buffer was exchanged for 50 mM HEPES and 10 mM MgCl₂ using 10 mL Zeba™ Spin Desalting Columns (Thermo Scientific). The purified proteins were analyzed with native and SDS-PAGE and quantified with Qubit™ Protein Assay Kit (Life Technologies).

**Protein gel electrophoresis.** For both native and SDS-PAGE, 4-15% Mini-PROTEAN TGX gels (Bio-Rad) were used. 10x running buffer consisted of 250 mM Tris, 1.92 M glycine for native PAGE and 1% (w/v) SDS was added to the SDS-PAGE running buffer. 2x native sample buffer consisted of 62.5 mM Tris-HCl, 40% (v/v) glycerol, 0.01% bromophenol blue. As a denaturating sample buffer, 2x Laemmli buffer with 10% 2-mercaptoethanol was used. Gels were run at 100-120 V for 60 min in a Mini-PROTEAN Tetra Cell (Bio-Rad) and stained with ReadyBlue™ Protein Gel Stain (Sigma-Aldrich) for 1 h. Gel densitometry analysis was performed in ImageJ 1.50e.

**Thermal stability assays.** Melt curves of purified wild-type and phosphomutant enzymes were recorded on QuantStudio 3 Real-Time PCR System (Applied Biosystems) using the QuantStudio™ Design & Analysis Software v1.4.3. Purified proteins in HEPES buffer at pH 7.5 were mixed with buffer and dye from Protein Thermal Shift™ Dye Kit (Applied Biosystems), and protein melt curves were recorded with settings specified in the kit protocol. Protein $T_m$ was determined as a maximum of the melt curve derivative. For each enzyme, melt curves were recorded in six replicates.

**In vitro enzyme assays.** In vitro activity of purified enzymes was determined by monitoring substrate decrease or product formation using an Infinite® M Nano spectrophotometer (Tecan) and Tecan i-control 3.9.1.0 software. The assays were performed at 37 °C at least in four replicates. Pta activity was monitored by measuring the formation of acetyl-CoA at 233 nm, PykF—decrease of

phosphoenolpyruvate at 240 nm, Gnd— formation of NADPH at 340 nm. TpiA reaction was coupled to GapA so that the amount of GapA is not limiting, and the activity of TpiA could be monitored by NADH formation at 340 nm. All reactions were prepared in 100 mM HEPES buffer at pH 7.5 and the specific components were: for Gnd—10 mM MgCl₂, 1 mM NADP+, and 0.1, 1 or 5 mM 6-phosphogluconate; for PykF—10 mM MgCl₂, 10 mM KCl, 1 mM fructose 1,6-bisphosphate, 1 mM ADP, 2 mM phosphoenolpyruvate; for Pta—10 mM MgCl₂, 1 mM coenzyme A, 5 mM acetyl-phosphate; for TpiA—GapA, 10 mM Na₂HPO₄, 1 mM NAD+, and 1 or 10 mM dihydroxyacetone phosphate. Initial reaction rates were calculated as the slope of linear substrate decrease/product formation over time, and values were normalized by the protein concentration. The initial reaction rates of mutants were divided by the wild-type values to obtain relative enzymatic activities.

**Intact protein mass spectrometry for PTM stoichiometry and localization.** For each enzyme, 10 μg of protein was separately buffer exchanged against milli-Q water on an Amicon 10 kDa molecular weight cutoff filter at 4 °C (Merck Millipore). A total of 1 μg of protein was then loaded for liquid chromatography (LC)-MS/MS analysis using a Vanquish UHPLC liquid chromatography system with the autosampler set to 4 °C. Protein was resolved across a linear gradient of buffer B (99.9% acetonitrile/0.1% formic acid) against buffer A (99.9% H₂O/0.1% formic acid) at 150 μL/minute on a MabPac 1×150 mm column (Thermo) with the column heater set to 50 °C. The LC was run online to a Thermo Q-Exactive HF mass spectrometer using a HESI source operated in positive mode, with spray voltage set to +3.5 kV, a capillary temperature of 310 °C, the source ion funnel radio frequency (RF) level set to 40%, and sheath gas flow rate set to 10. Full MS data was acquired to examine sample purity and intact protein mass. Full profile data was collected with 10 microscans at a resolution of 7,500 (at 200 m/z) from 500-2000 m/z with a target automatic gain control (AGC) of 3*10⁶ charges and a maximum injection time (IT) of 100 msec. MS2 data was acquired with a targeted selected ion monitoring scan (tSIM) with a 50 m/z isolation window around the 30+ charge state of each enzyme. To increase sequence coverage, each enzyme was subjected to higher energy collisional dissociation with normalized collisional energy values ranging from 10-23. Given the limits of instrument control software, the default precursor charge state was set to 24 + . MS2 data was acquired at 120,000 resolution (at 200 m/z) with 4 microscans with a target AGC of 3×10⁶ charges and a maximum injection time of 200 ms. Summed full scan spectra were deconvoluted to neutral masses using UniDec 4.4.1[69]. MS2 data was deconvoluted using the Xtract algorithm as part of FreeStyle 1.7 (Thermo Fisher). Fragment masses, sequence-level modifications, and PTMs were then assigned to primary protein sequences using ProSight Lite 1.4[70].

**Metabolic flux analysis.** ¹³C-labeling experiments were performed in 500 ml shake flasks containing 50 mL of M9 minimal medium with 3 g/L of either a mixture of 20% [U-¹³C] and 80% of natural abundance glucose, or 100% [1-¹³C]-labeled glucose. Aliquots of fractionally ¹³C-labeled biomass were withdrawn during mid-exponential growth and mass isotopomer pattern in proteinogenic amino acids were determined by GC-MS[49]. For data analysis, we used the Matlab-based software FiatFlux to determine ratios of converging fluxes and to combine this information with fluxes in and out of the cell and growth rates to obtain absolute flux estimations[50,71].

**Reporting summary.** Further information on research design is available in the Nature Research Reporting Summary linked to this article.

## Data availability

The mass spectrometry proteomics data generated in this study have been deposited in the ProteomeXchange Consortium via the PRIDE[72] partner repository under accession code PXD027243. The metabolomics data generated in this study have been deposited in the MassIVE database under accession code MSV000087795. Other data are available in the Supplementary Data and Supplementary Information. Source data are provided with this paper.

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

## Acknowledgements

The authors would like to thank Dr. Tobias Fuhrer for assistance with flux analysis and Dr. Ludovic Gillet for providing phosphoproteomics expertise. H.H.W. acknowledges partial funding support from NIH (1R01AI132403, 1R21AI146817), Burroughs Wellcome Fund (1016691), and NSF (MCB-2032259).

## Author contributions

U.S., E.S., and Z.R.N. designed the study. E.S., N.J., J.P., and H.H.W. prepared the *E. coli* phosphosite mutants. E.S., Z.R.N., and A.K. carried out growth and metabolomics experiments and analyzed the data. E.S. and C.H.G. performed in vitro experiments. P.F.D. analyzed the PTM status of purified enzymes. E.S. performed flux analysis. U.S., E.S., and Z.R.N. wrote the manuscript. All authors reviewed and approved the final manuscript.

## Competing interests

The authors declare no competing interests. H.H.W. is a member of the scientific advisory board and equity holder of SNIPR Biome who is not involved with this work.
