## [Peer Review File · Nature Communications]

Reviewer comments, first round:

-

Reviewer #1 (Remarks to the Author):

In this manuscript, focusing on the phosphorylation of enzymes in central metabolism pathways, the authors analyzed the effects of Ala and Glu substitutions on the growth rate of cells, and the effects of these mutations on the metabolome profiles. As a result, they found that 85% of the investigated phosphosites showed some functional responses. In addition, the authors performed an in vitro thermal shift assay to evaluate the thermal stability of the enzymes, in vitro evaluation of the enzyme activity, and flux analysis for validation.

Overall, the experiments were thorough and the data was processed appropriately. However, due to the large-scale analysis, detailed functional analysis of individual molecules has not yet been achieved.

Since phosphorylation stoichiometry in bacteria proteomes is known to be very low. It has been reported that on average less than 1% of the molecules are phosphorylated for bacteria, whereas 80% are phosphorylated for mammalian proteins (Anal. Chem. 2006, 78, 6, 1987). If this is true, how can they explain the results of the mutation effects on the combined metabolic and phenotypic changes? Is there any possibility to introduce the artificial perturbation on top of the phospho-mimetic or nonphosphorylatable mutation? It would be necessary to do more detailed functional analysis to show how the phosphorylation with low stoichiometry regulates the function in cells for a few enzymes.

Regarding the experimental details, especially metabolome analysis, the employed conditions including MS parameters should be described clearer since FIA-TOF analysis would have poorer power to separate metabolites, in comparison to GC/MS, LC/MS, LC/MS/MS or CE/MS. In addition, all of MS raw files should be deposited to public repositories.

Reviewer #2 (Remarks to the Author):

In their methodologically sound and well written manuscript Schastnaya et al. characterise in great detail a number of previously postulated bacterial phosphosites.

The authors aim at identifying the function of several such phosphosites of the STY type, mainly in enzymes of the central metabolism, by mutating the respective amino acids to either mimic a state of constant phosphorylation or render the position non-phosphorylatable.

They then analyse these mutants, employing in vivo and in vitro approaches enabling them to deduce functionality. For most of the enzymes under investigation the authors can convincingly pinpoint the observed effects to either stimulating or repressing effects of phosphorylation events, or non-specific denaturation of the protein in question due to inactivation by conformation changes, even if no obvious phenotype can be observed.

This manuscript deepens our understanding of PTM dependent regulation in bacteria and is of general interest to a broader readership. It should be published with some minor corrections.

Major points:

Serine can be subjected to other types of PTM. Hence phenotypic changes following replacement with alanine could be unrelated to phosphorylation (ref. 35).

Negative charge of substitute (-1) differs from that of phosphorylation (-1,5) and could cause effects not related to phosphorylation (ref. 36).

Minor points:

1. Abstract "P is arguably the most important post-translation modification in eukaryotes" reads

dogmatic. For example, there is a single hypusination side in the human proteome, and without it, no cell would survive. Saying "P is an essential and highly frequent modification..." would be more scientific.

Data and methodology:

2. The use of the metabolite extraction procedure could be better justified.
3. "Intact protein mass spectrometry for PTM stoichiometry and localization.",
 - a. Please explain better any need for the enrichment of phosphoproteins
 - b. Please explain better the need mAbs / ADCs
 - c. Please define polarity

While the paper is generally well written, it would benefit from some changes in wording, i.e.

Line 23 should read: "...fewer proteins were found to be phosphorylated..."

Line 28 should read: "...we found 44 of investigated 52 mutated phosphosites..."

Line 193 should read: "...Gnd S304 and PykF S198 do activate enzyme activity..."

Reviewer #3 (Remarks to the Author):

Schastnaya et al. performed an extensive analysis of regulation of enzyme activity by phosphorylation in *E. coli*. They mutated 52 phosphosites (on 23 proteins) reported in several phosphoproteomic studies and tested the effects of the corresponding phospho-mimetic, -ablative and KO mutants on growth in minimal medium with glucose and/or several other carbon sources (acetate, pyruvate, glycerol, fructose). The authors report that a majority of tested mutants showed a growth phenotype under at least one of two tested conditions. They coupled this analysis with metabolome profiling, showing phenotypes in metabolic changes and signatures for 38 phosphosites, including 14 that did not show a growth phenotype. Finally, they performed in vitro characterization of enzymatic activities of four enzymes that exhibited significantly reduced growth rate compared to WT. For two of these enzymes (TpiA and Pta), they could confirm direct inhibition by phosphorylation in vitro.

Overall, this is an important and carefully executed metabolomics study that points to functional relevance of an unprecedented number of bacterial phosphorylation events. As such, it will be a valuable resource for scientific community working on regulation of bacterial metabolism and of interest to the broad readership of Nature Communications. However, before publication, the authors should address four important points:

1) Since all growth phenotypes were reported as comparison to the WT, the authors should analyze and report the actual phosphorylation status (incl. occupancy, see below) of phosphosites on relevant enzymes in the WT. This can be done by a dedicated (untargeted or targeted) phosphoproteome measurement of the WT strains of several selected enzymes under exact conditions studied in this manuscript.

2) One major aspect of bacterial protein phosphorylation is low occupancy of phosphorylation events. Since the phospho-mimetic/ablative/KO mutants do not address this aspect, the authors should provide evidence, from the literature and direct measurements (see point 1), on the WT occupancy of relevant phosphosites from this study.

3) An in vitro measurement of the influence of the phosphosite occupancy on enzymatic activity (e.g. by mixing phosphomimetic and -ablative TpiA mutants in different ratios and measuring activity), would be a major addition to the manuscript.

4) An effort to model/predict the effect of substoichiometric occupancy of selected key phosphosites such as on Pta or TpiA on metabolic flux would be very beneficial.

Reviewer #1 (Remarks to the Author):

In this manuscript, focusing on the phosphorylation of enzymes in central metabolism pathways, the authors analyzed the effects of Ala and Glu substitutions on the growth rate of cells, and the effects of these mutations on the metabolome profiles. As a result, they found that 85% of the investigated phosphosites showed some functional responses. In addition, the authors performed an *in vitro* thermal shift assay to evaluate the thermal stability of the enzymes, *in vitro* evaluation of the enzyme activity, and flux analysis for validation.

Overall, the experiments were thorough and the data was processed appropriately. However, due to the large-scale analysis, detailed functional analysis of individual molecules has not yet been achieved.

Since phosphorylation stoichiometry in bacteria proteomes is known to be very low. It has been reported that on average less than 1% of the molecules are phosphorylated for bacteria, whereas 80% are phosphorylated for mammalian proteins (Anal. Chem. 2006, 78, 6, 1987). If this is true, how can they explain the results of the mutation effects on the combined metabolic and phenotypic changes?

We thank the reviewer for the positive assessment of our manuscript and raising the important question of the phosphorylation stoichiometry. Indeed, some PTMs in bacteria have very low stoichiometry. However, so far the only systematic attempt to quantify the phosphorylation stoichiometry in *E. coli* growing in minimal medium with glucose, the condition we used in this study, reported that the median occupancy of phosphorylation sites was 6.7% and 11.5% during mid-exponential and stationary growth phases, respectively (Soares et al., 2013). The phosphosite with the highest reported occupancy was FbaB S345 (67%), and indeed we saw a decrease in growth rate when we abolished phosphorylation at this site and no change in growth rate compared to the wild-type when we mimicked phosphorylation. We now point out explicitly in results and discussion that a 100% stoichiometry introduced by the phospho-mimicking mutation does not reflect the wild-type phosphosites occupancy but rather is a genetic loss/gain of function experiment (lines 93-97, 273-275). Generally, phospho-mutations are pertinent to assess the overall functionality of a phosphosite, but do not necessarily reflect the site occupancy *in vivo*, which we now also point out in the discussion. Nevertheless, for nine of our phosphosites the stoichiometry is available under the here investigated conditions, and it is much higher than 1%, as we report now in a new column of Supplementary Table 1. We respectfully disagree though that functional analysis has not yet been achieved. In several of the seemingly most relevant cases we provide functional data at the level of *in vitro* activities, protein folding, and *in vivo* pathway usage.

Is there any possibility to introduce the artificial perturbation on top of the phospho-mimetic or nonphosphorylatable mutation? It would be necessary to do more detailed functional analysis to show how the phosphorylation with low stoichiometry regulates the function in cells for a few enzymes.

We absolutely agree with the reviewer that titrating phosphorylation stoichiometry could be very useful to quantify functionality of individual phosphosites. Unfortunately, no such methods are currently available for *in vivo* experiments. Instead, following the advice of reviewer #3 (see point 3), we now performed an *in vitro* titration experiment that demonstrated that a site occupancy similar to the one reported *in vivo* has a significant activity reduction also *in vitro* (Supplementary Figure 1).

Regarding the experimental details, especially metabolome analysis, the employed conditions including MS parameters should be described clearer since FIA-TOF analysis would have poorer

power to separate metabolites, in comparison to GC/MS, LC/MS, LC/MS/MS or CE/MS. In addition, all of MS raw files should be deposited to public repositories.

We added the MS parameters to the corresponding Methods section (lines 369-377): “A sample volume of 5 μ L was injected into a constant flow of isopropanol/water (60:40, v/v) buffered with 5 mM ammonium carbonate (pH 9), containing 3-Amino-1-propanesulfonic acid (138.0230374 m/z, Sigma Aldrich) and hexakis(1H,1H,3H-tetrafluoropropoxy)phosphazine (940.0003763 m/z, HP-0921, Agilent Technologies) for online mass axis correction. The ion source parameters were set as follows: 225°C source temperature, 11 L/min drying gas, 20 psig nebulizer pressure, and TOF settings as follows: 350 V fragmentor voltage, 750 V octopole voltage. Mass spectra were recorded in negative ionization mode within a mass/charge ratio range of 50 – 1000 m/z using the highest resolving power (4 GHz HiRes) with an acquisition rate of 1.4 spectra per second”. We also deposited all raw MS files in public repositories and listed the identifiers in the Data availability statement (lines 468-471). Once the manuscript is published, the datasets will become publicly available.

Reviewer #2 (Remarks to the Author):

In their methodologically sound and well written manuscript Schastnaya et al. characterise in great detail a number of previously postulated bacterial phosphosites.

The authors aim at identifying the function of several such phosphosites of the STY type, mainly in enzymes of the central metabolism, by mutating the respective amino acids to either mimic a state of constant phosphorylation or render the position non-phosphorylatable.

They then analyse these mutants, employing in vivo and in vitro approaches enabling them to deduce functionality. For most of the enzymes under investigation the authors can convincingly pinpoint the observed effects to either stimulating or repressing effects of phosphorylation events, or non-specific denaturation of the protein in question due to inactivation by conformation changes, even if no obvious phenotype can be observed.

This manuscript deepens our understanding of PTM dependent regulation in bacteria and is of general interest to a broader readership. It should be published with some minor corrections.

Thanks for the appreciation!

Major points:

Serine can be subjected to other types of PTM. Hence phenotypic changes following replacement with alanine could be unrelated to phosphorylation (ref. 35).

Negative charge of substitute (-1) differs from that of phosphorylation (-1,5) and could cause effects not related to phosphorylation (ref. 36).

In principle the reviewer is correct, but we fail to see why this would be a major concern against publishing this work, as it pertains to any study using PTM-modifying mutations. To point this potential confounding factor out, we now added the below sentence to the discussion (lines 268-270).

“Since serine and threonine are also subject to other modifications, we cannot exclude that some phenotypic changes following replacement with alanine might at least in part be influenced by other PTMs.”

Minor points:

1. Abstract “P is arguably the most important post-translation modification in eukaryotes” reads dogmatic. For example, there is a single hypusination side in the human proteome, and without it, no cell would survive. Saying “P is an essential and highly frequent modification...” would be more scientific.

We agree with the reviewer and changed the wording in the abstract according to the reviewer’s suggestion (line 21).

Data and methodology:

2. The use of the metabolite extraction procedure could be better justified.

The employed extraction procedure has been optimized over the years in our lab for the here used FIA-based mass spectrometry. In several places we refer to the original method establishment papers, but prefer to not overburden the present manuscript with non-essential technical detail.

3. “Intact protein mass spectrometry for PTM stoichiometry and localization.”,

a. Please explain better any need for the enrichment of phosphoproteins

We thank the reviewer for their comments to improve the readability of our manuscript and reproducibility of our methods. In our manuscript there was no need for enrichment of phosphoproteins, and our methods section accurately reflects the exact methods and analyses that were performed. For all proteins investigated in this manuscript, we first introduced the initially identified phosphosite mutation into His-tagged proteins expressed from overexpression plasmids, to then use His GraviTrap TALON columns for affinity purification, which is entirely different from typical enrichment methods used for proteomics of the whole cell samples. This is described in the “Expression and purification of phosphomutant enzymes” section of the experimental methods.

b. Please explain better the need mAbs / ADCs

We did not use any monoclonal antibodies or antibody-drug conjugates, and also did not mention them in the manuscript. The purification of the investigated enzymes was achieved by His-tag purification.

c. Please define polarity

We have updated the methods section to more explicitly state that proteomics data was collected in positive electrospray ionization mode (line 443). The vast majority of proteomics studies and targeted mass spectrometry-based studies of proteins are performed in positive mode, as summarized in this great review article of the mechanism of ESI for whole proteins (PMID: 23134552). This review shows the mechanism of positive polarity ESI for whole, globular proteins.

While the paper is generally well written, it would benefit from some changes in wording, i.e. Line 23 should read: “....fewer proteins were found to be phosphorylated...”

We changed the sentence according to the reviewer's suggestion.

Line 28 should read: "...we found 44 of investigated 52 mutated phosphosites..."

We thank the reviewer for carefully reading our manuscript and putting effort into suggesting better wording options. In this specific case, we prefer to keep the original phrasing "...we found 44 of the 52 mutated phosphosites to be functional...", as saying both "investigated" and "mutated" seems excessive.

Line 193 should read: "...Gnd S304 and PykF S198 do activate enzyme activity..."

We changed the wording, and now it says "...phosphorylation of Gnd S304 and PykF S198 activates enzyme activity...".

Reviewer #3 (Remarks to the Author):

Schastnaya et al. performed an extensive analysis of regulation of enzyme activity by phosphorylation in *E. coli*. They mutated 52 phosphosites (on 23 proteins) reported in several phosphoproteomic studies and tested the effects of the corresponding phospho-mimetic, -ablative and KO mutants on growth in minimal medium with glucose and/or several other carbon sources (acetate, pyruvate, glycerol, fructose). The authors report that a majority of tested mutants showed a growth phenotype under at least one of two tested conditions. They coupled this analysis with metabolome profiling, showing phenotypes in metabolic changes and signatures for 38 phosphosites, including 14 that did not show a growth phenotype. Finally, they performed *in vitro* characterization of enzymatic activities of four enzymes that exhibited significantly reduced growth rate compared to WT. For two of these enzymes (TpiA and Pta), they could confirm direct inhibition by phosphorylation *in vitro*.

Overall, this is an important and carefully executed metabolomics study that points to functional relevance of an unprecedented number of bacterial phosphorylation events. As such, it will be a valuable resource for scientific community working on regulation of bacterial metabolism and of interest to the broad readership of Nature Communications. However, before publication, the authors should address four important points:

1) Since all growth phenotypes were reported as comparison to the WT, the authors should analyze and report the actual phosphorylation status (incl. occupancy, see below) of phosphosites on relevant enzymes in the WT. This can be done by a dedicated (untargeted or targeted) phosphoproteome measurement of the WT strains of several selected enzymes under exact conditions studied in this manuscript.

We apologize if it wasn't clear from the text (and we further clarified it) that all mutated phosphosites were already reported in proteomics studies to be phosphorylated *in vivo* in the wild-type strain. Although we agree that knowing the precise site occupancy would be valuable, this is technically challenging even for dedicated proteomics labs. As a consequence, the stoichiometry of phosphorylation was so far identified for less than 100 of the around 3000 identified phosphosites in *E. coli*. Even for the case of isocitrate dehydrogenase inactivation via phosphorylation during growth on acetate, known since 1980s, we are not aware of a study that reported the actual occupancy of this modification *in vivo*. Since even top proteomics studies rarely report such values, providing

stoichiometry data far extends the scope of this functional metabolomics study. We are planning to pursue this question in the future but it will have to be subject to separate study. Nevertheless, as mentioned in response to Reviewer #1, the *in vivo* stoichiometry during growth in minimal medium with glucose (the exact condition studied here) is known for nine out of 52 investigated phosphosites, and the highest reported occupancy of phosphorylation site was reported for FbaB S345 (67%) (Soares et al., 2013).

2) One major aspect of bacterial protein phosphorylation is low occupancy of phosphorylation events. Since the phospho-mimetic/ablative/KO mutants do not address this aspect, the authors should provide evidence, from the literature and direct measurements (see point 1), on the WT occupancy of relevant phosphosites from this study.

We agree and now provide these data from the literature in Supplementary Table 1. See also our responses to Reviewer #1 and the question above.

3) An *in vitro* measurement of the influence of the phosphosite occupancy on enzymatic activity (e.g. by mixing phosphomimetic and –ablative TpiA mutants in different ratios and measuring activity), would be a major addition to the manuscript.

Thanks for this valuable suggestion. In addition to the enzymatic activities of unphosphorylated TpiA wild-type and phosphomimicking mutants currently reported in the manuscript, we now added the measurements suggested by the reviewer. The results are shown in the Supplementary Figure 1 and Supplementary Table 7. As the occupancy of S177 phosphorylation site was recently suggested to be around 15% during growth in minimal medium with glucose (Sultan et al., 2021), we mixed the unphosphorylated wild-type enzyme with phosphomimicking TpiA S177E at 85% wt/15% mutant ratio to approximate the *in vivo* stoichiometry. Additionally, we measured the activity of the 50% wt/50% TpiA S177E mix. As expected, the more phosphomimicking mutant there is in the mix, the lower the enzymatic activity. The activity decreased from 0% to 100% phosphorylation and can be approximated with a linear fit with $R^2=0.97$. We also added an additional sentence to the text (lines 204-207): “The site occupancy of S177 and T179 was recently reported to be around 15% during growth on glucose (Sultan et al., 2021), a percentage that significantly decreases TpiA *in vitro* activity (Supplementary Figure 1) and hence glycolytic flux”.

4) An effort to model/predict the effect of substoichiometric occupancy of selected key phosphosites such as on Pta or TpiA on metabolic flux would be very beneficial.

As much as we agree with the reviewer, this would clearly go beyond the scope of the present study that aimed to identify phosphosite functionality. At this point such a model would, for our taste, also involve too much hand waving because the stoichiometry of phosphorylation is not available. Now that our study demonstrated which sites in central metabolism have functional consequences for enzyme activity and we even know which fluxes are potentially regulated by it, we have set the stage for such a more quantitative, potentially model-based analysis. We are actually planning to pursue this research, but it will require extensive new data to obtain site occupancy.

Reviewer comments, second round:

-

Reviewer #1 (Remarks to the Author):

This is a revised version of NCOMMS-21-16602-T. The authors have responded appropriately to my concerns and comments, presenting new data. I fully understand that it is technically difficult to measure the phosphorylation stoichiometry on a large scale, especially in bacteria. On the other hand, given the difficulties of bacterial phosphoproteomics, it seems certain that phosphorylation sites with 10-60% stoichiometry, such as those shown by the authors, are rare, and that most phosphorylation sites have much lower stoichiometry. At this stage, it might be publishable. Later, this mystery will be solved with the introduction of new technology.

Reviewer #3 (Remarks to the Author):

Schastnaya et al. submitted a revised version of their manuscript on regulation of enzyme activity by phosphorylation in *E. coli*. They addressed one of my suggestions (mixing phosphomimetic and -ablative TpiA mutants in different ratios and measuring activity), which led to interesting results and a new supplementary figure. Overall, the manuscript has improved in sense that the phosphorylation site occupancy is now more thoroughly addressed in the text. My first comment was misunderstood - already a simple phosphoproteomics analysis would be beneficial to confirm whether the phosphorylation events addressed in the experiments are actually present in the WT (they are obviously already reported in the literature, but specific experimental conditions, growth phase, etc. may influence their occurrence). However, I agree that such an experiment would likely be incomplete (and would have to be done on exactly the same sample), so I do not see this as an obstacle for publication. I see that most of the other reviewers' comments were also addressed and I endorse the publication of the manuscript in *Nature Communications*.

Reviewer #1 (Remarks to the Author):

This is a revised version of NCOMMS-21-16602-T. The authors have responded appropriately to my concerns and comments, presenting new data. I fully understand that it is technically difficult to measure the phosphorylation stoichiometry on a large scale, especially in bacteria. On the other hand, given the difficulties of bacterial phosphoproteomics, it seems certain that phosphorylation sites with 10-60% stoichiometry, such as those shown by the authors, are rare, and that most phosphorylation sites have much lower stoichiometry. At this stage, it might be publishable. Later, this mystery will be solved with the introduction of new technology.

We thank the reviewer for their helpful comments and approval of the final version of our manuscript.

Reviewer #3 (Remarks to the Author):

Schastnaya et al. submitted a revised version of their manuscript on regulation of enzyme activity by phosphorylation in *E. coli*. They addressed one of my suggestions (mixing phosphomimetic and –ablative TpiA mutants in different ratios and measuring activity), which led to interesting results and a new supplementary figure. Overall, the manuscript has improved in sense that the phosphorylation site occupancy is now more thoroughly addressed in the text. My first comment was misunderstood – already a simple phosphoproteomics analysis would be beneficial to confirm whether the phosphorylation events addressed in the experiments are actually present in the WT (they are obviously already reported in the literature, but specific experimental conditions, growth phase, etc. may influence their occurrence). However, I agree that such an experiment would likely be incomplete (and would have to be done on exactly the same sample), so I do not see this as an obstacle for publication. I see that most of the other reviewers' comments were also addressed and I endorse the publication of the manuscript in Nature Communications.

We would like to thank the reviewer for approving the publication and for suggesting to investigate the effect of TpiA phosphosite occupancy on its enzymatic activity, which added an important result to the paper.